# Two Genetic Mechanisms in Two Siblings with Intellectual Disability, Autism Spectrum Disorder, and Psychosis

**DOI:** 10.3390/jpm12061013

**Published:** 2022-06-20

**Authors:** Yu-Shu Huang, Ting-Hsuan Fang, Belle Kung, Chia-Hsiang Chen

**Affiliations:** 1Department of Psychiatry, Chang Gung Memorial Hospital-Linkou, Taoyuan 33343, Taiwan; yushuhuang1212@gmail.com; 2Department of Psychiatry, College of Medicine, Chang Gung University, Taoyuan 33302, Taiwan; 3Department and Institute of Biomedical Sciences, Chang Gung University, Taoyuan 33302, Taiwan; genie.cgu@gmail.com (T.-H.F.); isabella900217@gmail.com (B.K.)

**Keywords:** intellectual disability, autism spectrum disorder, developmental disorder, psychosis, genetics, molecular diagnosis

## Abstract

Intellectual disability (ID) and autism spectrum disorder (ASD) are complex neurodevelopmental disorders with high heritability. To search for the genetic deficits in two siblings affected with ID and ASD in a family, we first performed a genome-wide copy number variation (CNV) analysis using chromosomal microarray analysis (CMA). We found a 3.7 Mb microdeletion at 22q13.3 in the younger sister. This de novo microdeletion resulted in the haploinsufficiency of *SHANK3* and several nearby genes involved in neurodevelopment disorders. Hence, she was diagnosed with Phelan–McDermid syndrome (PMS, OMIM#606232). We further performed whole-genome sequencing (WGS) analysis in this family. We did not detect pathogenic mutations with significant impacts on the phenotypes of the elder brother. Instead, we identified several rare, likely pathogenic variants in seven genes implicated in neurodevelopmental disorders: *KLHL17*, *TDO2*, *TRRAP*, *EIF3F*, *ATP10A*, *DICER1*, and *CDH15*. These variants were transmitted from his unaffected parents, indicating these variants have only moderate clinical effects. We propose that these variants worked together and led to the clinical phenotypes in the elder brother. We also suggest that the combination of multiple genes with moderate effects is part of the genetic mechanism of neurodevelopmental disorders.

## 1. Introduction

Intellectual disability (ID) and autism spectrum disorder (ASD) are common neurodevelopmental disorders. The prevalence of ID is around 1–3% in the general population worldwide [1,2,3], while the global prevalence of ASD is approximately 1% [4]. ID and ASD are complex disorders with high clinical heterogeneity and etiological complexity. Approximately 33% of patients with ASD are comorbid with ID [4], while ID often co-occurs with behavioral and psychiatric conditions [5,6,7]. ID and ASD have high genetic components in their etiology, and the genetic deficits of ID and ASD are highly heterogeneous, ranging from chromosomal abnormalities and copy number variations of genomic DNA (CNV) to small insertions and deletions (indel) and single nucleotide variants (SNV) [8,9]. More than 1000 genes are associated with ASD [10] and intellectual disability [11], and there are significant overlaps in pathogenic genes between ID and ASD, suggesting shared pathogenesis mechanisms between ID and ASD. Further, most pathogenic mutations associated with ID and ASD are individualized and personalized. Each affected patient and family have different pathogenic mutations. Identifying the genetic deficits in patients with ID and ASD is essential to understanding their pathophysiology, which could help with the clinical care and provide helpful information for counseling. However, it is challenging to establish the personalized molecular diagnosis for affected patients.

The advent of chromosomal microarray analysis (CMA) and next-generation sequencing (NGS) technology has significantly improved the diagnostic yield in patients with ID and ASD. CMA is a molecular genetic tool that can identify the location and the size of copy number variations of genomic DNA (CNV) with better resolution and accuracy than conventional karyotyping [12]. Accumulating studies indicate that CMA improves the genetic diagnostic rate of ID, ASD, and other neurodevelopmental disorders. Hence, the International Standards for Cytogenomic Array (ISCA) Consortium published a consensus statement and recommended using chromosome microarray as the first-tier genetic test for developmental disabilities and congenital abnormalities [13]. The consensus statement gained support from several studies [14,15,16,17], including our study of ASD [18].

Next-generation sequencing (NGS) is a massively parallel sequencing technology that can efficiently determine the genome sequences from organisms [19,20]. It can detect small indels and SNVs at a genome-wide level [21]. NGS has helped identify many genetic mutations associated with ID [22,23], ASD [24,25], and psychiatric disorders [26,27,28]. Some studies suggested using NGS as a first-tier genetic test for neurodevelopmental disorders [29,30]. Our group also identified several rare genetic mutations associated with ID and psychiatric conditions using NGS [31,32,33]. Our studies support that NGS helps establish a personalized molecular diagnosis for neurodevelopmental and psychiatric disorders [34].

In our series of molecular genetic studies of psychiatric disorders, we recruited singleton or multiplex families and searched for their genetic underpinnings using systematic genetic approaches, including conventional cytogenetic analysis, CMA, and NGS. This study searched for the genetic deficits in a family with two siblings affected with ID and ASD. Here, we report on our clinical and genetic studies of this family.

## 2. Materials and Methods

### 2.1. Subjects

Singleton and multiplex families diagnosed with neurodevelopmental disorders were recruited into our molecular genetic study series from Chang Gung Memorial Hospital-Linkou, Taoyuan, Taiwan. The Review Board of Chang Gung Memorial Hospital-Linkou approved the study with approval number 201801385A3. After we fully explained this study, each subject or their guardians signed the informed consent. We collected clinical information through interviews and reviews of medical records. The psychiatric diagnoses followed the criteria of the DSM-5 (Diagnostic and Statistical Manual of Mental Disorders, 5th edition). Genomic DNAs were prepared from each participant using the Smart Genomic DNA Extraction kit (Intelligent Biomedicine, Taipei, Taiwan).

### 2.2. Genome-Wide CNV Analysis

To detect CNV at a genome-wide level, we used the CytoScan HD Array (Affymetrix Inc., Santa Clara, CA, USA) platform. The experiments were performed at the Genomic Medicine Core Laboratory of Chang Gung Memorial Hospital-Linkou (Taoyuan, Taiwan). We used the Chromosomal Analysis Suite Version 3.3.0.139 (r10838) (Affymetrix Inc., Santa Clara, CA, USA) to analyze the data. Gain and loss of CNVs were detected at the resolution of 50 probes and 100 kb. The genomic coordinates of CNVs followed the human genome sequences version GRCh37/hg19. The interpretation of the clinical significance of CNV followed the “Technical standards for the interpretation and reporting of constitutional copy-number variants: a joint consensus recommendation of the American College of Medical Genetics and Genomics (ACMG, Bethesda, MD, USA) and the Clinical Genome Resource (ClinGen, Bethesda, MD, USA)” [35].

### 2.3. Real-Time Quantitative PCR (RT-qPCR)

We used real-time quantitative polymerase chain reaction (RT-qPCR) as a complementary method to verify the authenticity of CNV detected by the CytoScan HD array. In brief, we designed a primer pair (forward 5′-GGGTGGGGGCATTTTCTCTACCTT-3; reverse: 5′-GAGGCAGGAGGGAACCTCAGGA-3′) to obtain an amplicon of the *SHANK3* gene as the target gene. We also used another primer pair (forward 5′-CCGTGAACAGGTGAACAGCATTC-3′; reverse 5′-GCCTCTGCCTTACCTTTGTGTTT-3′) to obtain an amplicon of the *VIPR2* gene as the endogenous reference. The comparative ddCt method was used for the analysis of the RT-qPCR data. A dCt was first obtained by subtracting *VIPR2* Ct from *SHANK3* Ct. Then, the Ct of the tested subject was normalized to the control subject to obtain ddCt. The relative fold change to a normal subject was determined as 2^−ddCt^. The RT-PCR experiments were performed using the StepOnePlus machine (Applied Biosystems, Foster City, CA, USA) with the SYBR green method following the manufacturer’s instructions.

### 2.4. Whole-Genome Sequencing (WGS)

Whole-genome sequencing (WGS) was performed using the Illumina HighSeq2000 platform (Illumina, San Diego, CA, USA). After a quality check, the raw sequencing data were aligned to the human reference genome build hg19/GRch37. SAMtools and the Genome Analysis Tool Kit were used to refine the local alignment and generate a variant calling file (VCF). Variants were further annotated, filtered, and analyzed under different filtering criteria and inheritance models, including autosomal dominant, autosomal recessive, X-linked, and de novo mutation. The bioinformatics analysis of the NGS data in the family was performed using SeqLab software (ATgenomics, Taipei, Taiwan).

### 2.5. Sanger Sequencing

To verify the authenticity of mutations detected from WGS analysis, we first designed primer pairs to obtain amplicons covering the mutations by PCR. In brief, we performed 30 cycles of PCR in a 20 μL mixture containing 100 ng DNA, 1 μM of each primer, 1X buffer, 0.25 mM of dNTP, and 0.5 U of Power Taq polymerase (Genomics, New Taipei City, Taiwan) and with the annealing temperature of 63 °C. An aliquot of the amplicon was purified and subjected to Sanger sequencing using the BigDye Terminator kit v3.1 (Applied Biosystems, Foster City, CA, USA). We used the forward primer for sequencing.

### 2.6. Bioinformatics Analysis and Literature Review

This study defined a rare mutation with less than 1% minor allele frequency. The mutations identified in this study were checked in the dbSNP (https://www.ncbi.nlm.nih.gov/snp/, accessed on 8 June 2022) and the Taiwan Biobank (https://taiwanview.twbiobank.org.tw/index, accessed on 8 June 2022). The functional impacts of mutations were assessed using several online computer programs, including Polyphen-2 (http://genetics.bwh.harvard.edu/pph2/index.shtml, accessed on 8 June 2022), SIFT (https://sift.bii.a-star.edu.sg/, accessed on 8 June 2022), PROVEAN (http://provean.jcvi.org/index.php, accessed on 8 June 2022), and Mutation Taster (http://www.mutationtaster.org, accessed on 8 June 2022). The possible relevance of rare mutations to the pathogenesis of psychiatric disorders was evaluated by reviewing the literature in PubMed.

## 3. Results

### 3.1. Clinical Reports

We recruited a family with two affected siblings; the pedigree of this family is shown in Figure 1. The father and mother were 65 and 56 years old, respectively. They did not have a history of psychiatric disorders. The elder son was 30 years old and was born full-term without remarkable events. He was shy and had poor interpersonal and social interactions when he was a child. At 10 years old, he was diagnosed with moderate ID with ASD. He attended a special education program from elementary school to high school. At 23 years old, he started to manifest psychotic symptoms such as agitation, irritability, temper tantrums, aggressive behavior, self-talking, and self-laughing. Schizophrenia was added to his diagnosis. He received treatment with antipsychotics and mood stabilizers, but he responded poorly. He was admitted to the psychiatric ward several times due to unstable psychotic symptoms. At the age of 25, he received the Wechsler Adult Intelligent Scale-Fourth edition (WAIS-IV) test, which showed that he had a full-scale intelligence quotient (IQ) of 42, including Verbal Comprehension Index Scale of 50, Perceptual Reasoning Index Scale of 51, Working Memory Index Scale of 50, and Processing Index Scale of 50. At the age of 29, he received a course of electroconvulsive therapy (ECT) due to his intractable psychotic symptoms and catatonia, but he did not respond well to ECT. His mental conditions and social function were deteriorating gradually.

The younger sister was 26 years old. She was also born full-term without remarkable events. At the age of 4, she was diagnosed with severe ID and ASD due to global developmental delay, hypotonia, and lack of language development. She did not receive special education and stayed home with her parents. She had poor social function and self-care capability. At 21 years old, she manifested an unstable mood with irritability and agitation. She received antipsychotic treatment but responded poorly to pharmacotherapy. Currently, she still does not have verbal speech. Her mental condition remains the same. Unlike her elder brother, her mental symptoms did not worsen. Hence, she was not hospitalized and is presently under the custody of her parents.

### 3.2. Identification of a Pathogenic CNV in the Younger Sister

The genome-wide CNV analysis detected a 3753 kb subterminal interstitial deletion at chromosome 22q13.31 in the younger sister. The deletion started from nucleotide positions 47,445,140 to 51,197,725, which covered 52 genes. The readout of this microdeletion from the Chromosomal Analysis Suite Version 3.3.0.139 (r10838) (Affymetrix Inc., Santa Clara, CA, USA) is shown in Figure 2. The deleted genes are listed in Table 1. The CNV was considered pathogenic according to the “Technical standards for the interpretation and reporting of constitutional copy-number variants: a joint consensus recommendation of the American College of Medical Genetics and Genomics (ACMG) and the Clinical Genome Resource (ClinGen)” [35]. This pathogenic CNV was a de novo mutation that was not found in her parents or her elder brother. Real-time quantitative PCR verified the presence and the inheritance of this CNV in this family (Figure 3).

### 3.3. Detection of Multiple Rare Inherited Variants in the Elder Brother

WGS analysis did not detect pathogenic mutations associated with the clinical phenotypes of the elder brother under the models of dominant de novo mutation, autosomal recessive, and X-linked inheritance. Nevertheless, we identified seven rare inherited mutations with neuropsychiatric implications in the elder son. The authenticity and the origin of these rare mutations were verified by Sanger sequencing. The primer sequences, optimal annealing temperature, and amplicon size are listed in Table 2, and the chromatographs of the Sanger sequencing are shown in Figure 4. Five genetic variants were transmitted from his father, while two were inherited from his mother. The genetic information of these rare inherited mutations is summarized and listed in Table 3.

## 4. Discussion

We first detected a 3.7 Mb de novo interstitial microdeletion at chromosome 22q13.1–13.3 in the younger sister using CMA. Deletion at 22q13 is associated with Phelan–McDermid syndrome (PMS, OMIM#606232), a neurodevelopmental disorder characterized by moderate to profound intellectual disability, global developmental delay, delay or absence of speech development, ASD, and various behavioral problems [36,37,38,39,40]. Dysfunction of the *SHANK3* gene is thought to be the primary cause of PMS [36,38,41,42], as patients with *SHANK3* mutations manifest various neurodevelopmental and psychiatric conditions such as ID, ASD, and schizophrenia [43]. Several genes other than *SHANK3* deleted in the microdeletion region also contribute to PMS patients’ phenotypes [44,45]. The microdeletion in the younger sister resulted in the haploinsufficiency of 52 genes, including *SHANK3* and several nearby candidate genes. Hence, the young sister was diagnosed with PMS.

We did not detect any pathogenic CNV associated with the clinical phenotypes in the elder brother. We further analyzed the WGS data under different modes of inheritance, including de novo dominant mutation, recessive mutation, and X-linked hemizygous mutation, and we did not detect any pathogenic mutations fitting these models. The failure to detect major pathogenic mutations might be due to limitations of the WGS technology used in this study or mistakes in analyzing the WGS data. Nevertheless, we observed several rare variants in genes relevant to neuropsychiatric disorders in the patient. These rare, likely pathogenic variants were inherited from his unaffected parents.

From the father’s side, the patient inherited five missense mutations, including p.F268Y of *TDO2* (rs183229581), p.E2750D of *TRRAP* (rs55755466), p.P4S of *EIF3F* (rs367735033). p.R879H of *ATP10A* (rs184009994), and D98N of *CDH15* (rs149963083). The *TDO2* gene encodes the tryptophan 2,3-dioxygenase, a rate-limiting enzyme for the catabolism of tryptophan, the precursor of serotonin. Increased expression of *TDO2* was observed in the frontal cortex of patients with schizophrenia compared with that in controls [46], and in the postmortem of the anterior cingulate cortex, increased expression of mRNA and protein of *TDO2* was also detected in patients with schizophrenia and bipolar disorder [47]. *TDO2* gene was reported to be associated with Tourette syndrome [48] and autism [49]. *Tdo2*-knockout mice showed anxiolytic behaviors, increased adult neurogenesis [50], and enhanced exploratory behavior and cognitive function [51]. The TDO2 protein was also reported as a mediator of environmental factors associated with psychosis through epigenetic mechanisms [52].

The *TRRAP* gene encodes the transformation and transcription domain associated protein, belonging to the phosphoinositide 3-kinase-related kinases (*PIKK*) family. TRRAP is the standard component of many histone acetyltransferase complexes that are involved in the chromatin modification during the transcription, duplication, and repair of DNA [53]. TRRAP modulates gene transcription by regulating key transcription factors, such as E2F1, c-Myc, p53, and Sp1 [54,55]. *Trrap*-knockout mice were lethal [56]. Deleting *Trrap* in Purkinje neurons affected microtubule dynamics, resulting in neurodegeneration in old mice [54]. Notably, rare *TRRAP* mutations were found in some patients with schizophrenia [57], ID, and ASD [58,59,60].

The *EIF3F* gene encodes the eukaryotic translation initiation factor 3 subunit F, a component of the eukaryotic translation initiation factor 3 complex. The EIF3F protein is involved in IRES-dependent viral translational initiation, protein deubiquitination, and translational initiation. Mutations of the *EIF3F* gene were associated with recessive developmental disorders characterized by ID, epilepsy, behavioral problems, and various physical abnormalities [61,62].

The *ATP10A* gene encodes a putative ATPase phospholipid transportation protein 10A, which was mapped to 15q12, 200 kb distal to *UBE3A* [63]. *UBE3A* is a maternally expressed gene and is considered the candidate gene for the maternal duplication and deletion of 15q11–13. *ATP10A* is adjacent to *UBE3A* and is involved in the duplication and deletion of 15q11. Hence, it was also considered a candidate gene for 15q11–13 duplication and deletion syndromes [64]. Maternal duplication of 15q11–13 is associated with developmental delay, ASD, and seizure, while maternal deletion of 15q11–13 is associated with Angelman syndrome, a neurodevelopmental disorder characterized by ID, ASD, seizure, and other dysmorphic features. Several rare missense variants of the *ATP10A* gene were reported in patients with ASD, but their functional impacts are yet to be studied [64,65].

The *CDH15* gene encodes the cadherin 15 protein, a classic cadherin gene family member that belongs to the cadherin superfamily [66]. Mutations of multiple cadherin superfamily members are associated with neuropsychiatric disorders such as epilepsy, ID, ASD, bipolar disorder, and schizophrenia [67]. Rare pathogenic mutations of *CDH15* in patients with ID were reported. Bhalla and colleagues studied a translocation t(11; 16) (q24.2; q24) in a female patient with ID; they found that this translocation disrupted the *CDH15* and *KIRREL3* genes. They further screened for mutations of these two genes in a sample of 647 patients with idiopathic ID. They identified four heterozygous missense mutations of *CDH15* and three heterozygous missense mutations of *KIRREL3* in this sample. A functional study of three missense mutations of *CDH15* showed impaired reduction in cell adhesion, suggesting that they are pathogenic mutations [68].

From the mother’s side, the patient inherited the p.T366A of *KLHL17* (rs186429850) and the p.I37F of *DICER1* (rs772381832). The *KLHL17* gene is one member of the *KLHL* gene family, which encodes proteins that possess a BTB/POZ domain, a BACK domain, and five to six Kelch motifs [69]. The *KLHL17* gene is located at 1p36.33. Monosomy 1p36 deletion syndrome is the most common terminal deletion syndrome in humans, characterized by ID, developmental delay, seizures, dysmorphic features, and other physical abnormalities [70,71]. *KLHL17* was considered one of the candidate genes for 1p36 deletion syndrome [72]. The KLHL17 protein interacts with F-actin in the dendritic spines of neurons in the brain. Knockdown and knockout studies of *Klhl17* showed that KLHL17 modulated the remodeling of F-actin and contributed to the morphogenesis, maturation, and activity of neurons in the brain [73]. *Klhl17*-deficient mice (*Klhl17*+*/–)* showed hyperactivity and reduced social interaction, suggesting that dysfunction of *KLHL17* is associated with abnormal behaviors [73].

The *DICER1* gene encodes the dicer, ribonuclease III, which is responsible for generating RNA interference (RNAi), including small interference RNA (siRNA) and microRNA (miRNA) [74]. In addition to cleaving the double-strand RNA in the RNAi biogenesis, *DICER1* is involved in generating small RNAs and has non-endonuclease activities [75]. Mutations of *DICER1* are associated with susceptibility to various cancers, which was named DICER1 syndrome [76,77]. Some patients with *DICER1* mutations have additional phenotypes, such as global developmental delay, macrocephaly, ASD, and other physical abnormalities [78,79,80,81,82]. Notably, the increased expression of *DICER1* in the dorsolateral frontal cortex was reported in patients with schizophrenia [83,84]. An SNP (rs3742330) of *DICER1* was reported to be associated with Chinese schizophrenia [85], and a rare missense mutation of *DICER1* was detected in a Chinese patient with schizophrenia [86]. Hence, *DICER1* was considered a susceptibility gene for schizophrenia. Conditional knockout of *Dicer1* in excitatory forebrain neurons of mice led to microcephaly, reduced dendritic branch elaboration, and increased dendritic spine length in the brain [87].

Neuropsychiatric disorders are complex genetic disorders. With the progress of CMA and WGS analysis, accumulating evidence indicates that rare de novo mutations with high penetrance play an essential role in the genetic deficits of neuropsychiatric disorders. Nevertheless, a large part of neuropsychiatric patients’ genetic landscape is still missing. Several studies have indicated that oligemic involvement is part of the genetic landscape of neurodevelopmental disorders [32,88,89,90,91]. Notably, in a recent study, John and colleagues reported the detection of five rare heterozygous variants in a schizophrenia multiplex family. These likely pathogenic variants were inherited from unaffected parents. Hence, they proposed that these variants had cumulative and threshold effects on the development of schizophrenia [92]. Similar to this study, we detected seven rare, likely pathogenic variants in the elder brother in this family. These variants were transmitted from his unaffected parents, indicating that these variants have only moderate clinical impacts or reduced penetrance, but they may increase the carrier’s likelihood of developing neuropsychiatric disorders. When these variants happened to be present in the elder son, they might have interacted with each other and crossed the clinical threshold, resulting in the patient’s clinical phenotypes. We suggest that the oligogenic model of neuropsychiatric disorders might supplement the genetic architecture of neuropsychiatric disorders.

## 5. Conclusions

We discovered two distinct genetic mechanisms in two siblings affected by ASD, ID, and psychosis in one family. Our findings indicate that the genetic basis of neuropsychiatric disorders is complex even within a family. We also demonstrated the clinical utility of CMA and WGS in establishing the personalized molecular diagnosis for these two siblings. We hope that our findings can help improve the clinical care of these two siblings.

## Figures and Tables

**Figure 1 jpm-12-01013-f001:**
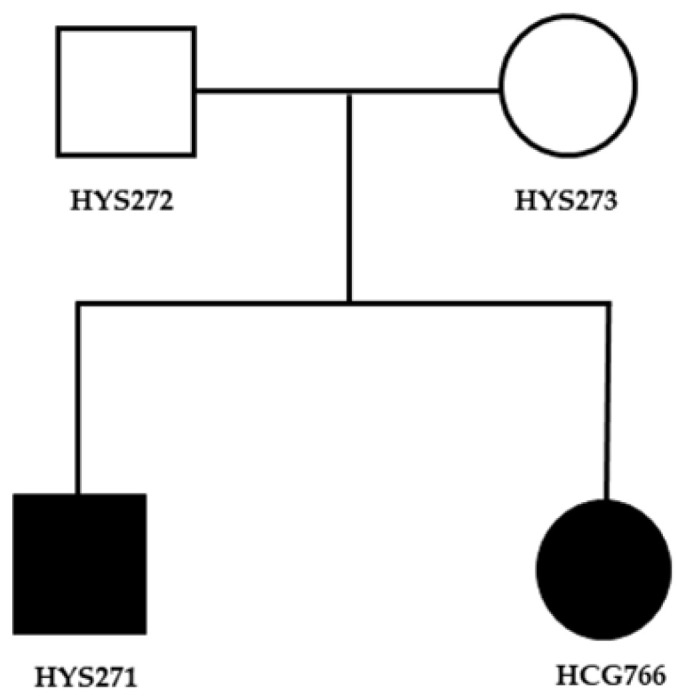
Pedigree of the family with two siblings affected with ID and ASD.

**Figure 2 jpm-12-01013-f002:**
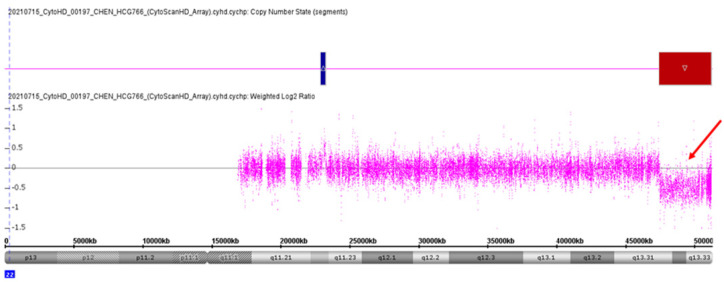
CMA revealed a 3.7 Mb microdeletion at 22q13.3 (red arrow) in the younger sister in this family.

**Figure 3 jpm-12-01013-f003:**
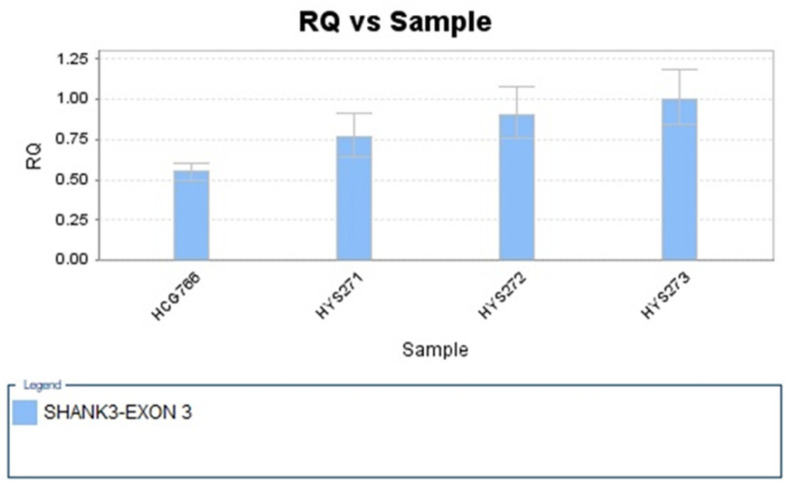
Real-time quantitative PCR showed the haploinsufficiency of an amplicon of *SHANK3* (exon3) in the younger sister (*HCG786*) but not in her elder brother (*HYS271*) or her parents (*HYS272* and *HYS273*).

**Figure 4 jpm-12-01013-f004:**
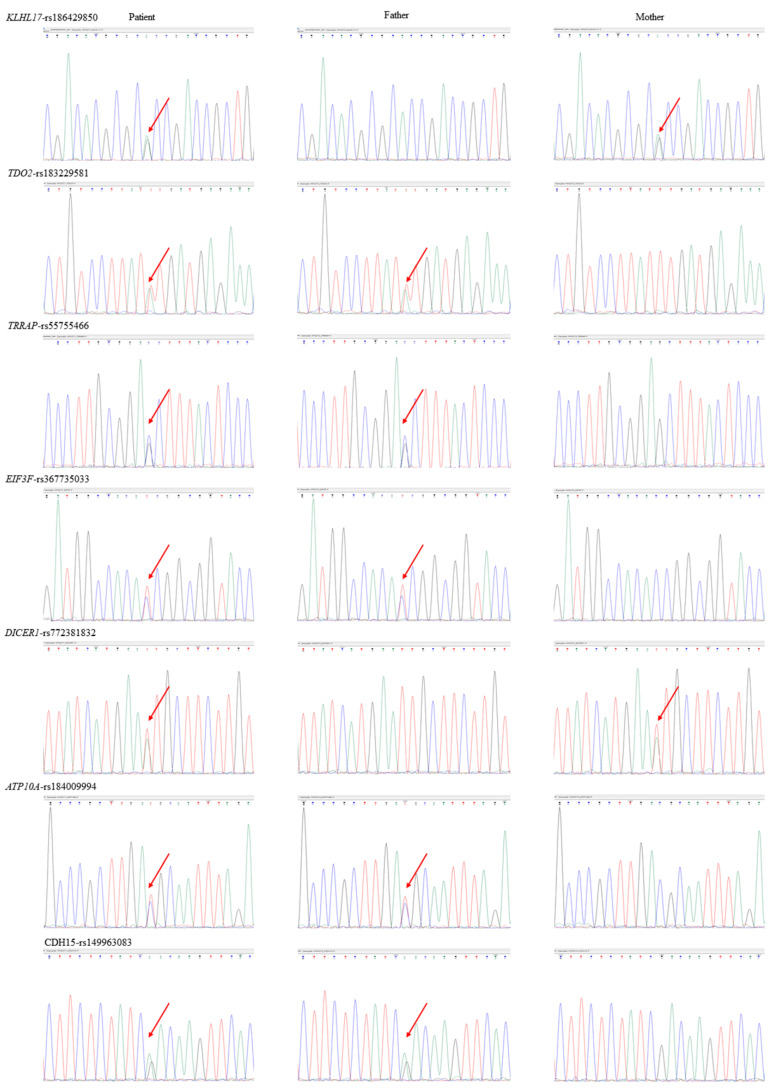
Sanger sequencing verified the authenticity and the origin of the seven inherited missense variants identified in the elder brother. Red arrows indicate the locations of the variants.

**Table 1 jpm-12-01013-t001:** List of genes deleted in the region of this microdeletion.

Cytogenetic location	22q13.31–13.33
Nucleotide Position	47445140-51197725
Size	3753 Kb
	*TBC1D22A*, *LOC339685*, *LINC01644*, *LINC00898*, *LOC284930*, *MIR3201*, *FAM19A5*, *LOC284933*, *MIR4535*, *LINC01310*, *C22orf34*, *MIR3667*, *BRD1* *, *ZBED4*, *ALG12* *, *CRELD2*, *PIM3*, *MIR6821*, *IL17REL*, *TTLL8*, *MLC1* *, *MOV10L1*, *PANX2*, *TRABD*, *SELENOO*, *TUBGCP6* *, *HDAC10*, *MAPK12*, *MAPK11*, *PLXNB2* *, *DENND6B*, *PPP6R2*, *SBF1* *, *ADM2*, *MIOX*, *LMF2*, *NCAPH2*, *SCO2* *, *TYMP* *, *ODF3B*, *KLHDC7B*, *SYCE3*, *CPT1B*, *CHKB-CPT1B* *, *CHKB*, *CHKB-AS1*, *MAPK8IP2* *, *ARSA* *, *SHANK3*, *LOC105373100*, *ACR*, *RPL23AP82*

* Candidate genes contribute to the patient’s phenotypes in addition to *SHANK3*.

**Table 2 jpm-12-01013-t002:** Primer sequences, optimal annealing temperature (Ta, °C), and size of PCR products for the verification of the mutations identified in this study using Sanger sequencing.

	Forward (5′-3′)	Reverse (5′-3′)	Size (bp)
*KLHL17*	CCCTCTTGCCCTGTGCCTTCTACT	CGGAATTAAGCCACTGCAGGTCAA	395
*TDO2*	CTCTCTCAGGACTATTAATGCC	AATCTGGGCATGGAAACCCGTT	338
*TRRAP*	GTGAGGGTGCGCCTCAGTTTGTTA	ACCCAAGACCGTCAGTGGTCTGAG	336
*EIF3F*	AGCAGAGCGCACAAATTCCAGAAG	AGGGTCTGAGGATGAGGCTGGAG	329
*DICER1*	GTGGGAGGCCTGAAAGGGTAAATG	CACTGGATGAATGAAAAGCCCTGC	262
*ATP10A*	GGAGCCACTTGAAACCCACCTACC	GTTCGCTCACACTGCTGTGCATTT	238
*CDH15*	GGAGACTTAGACCTGCCCTGCTGT	TAAGGGTGCCTGGATCTTGCAGTC	399

**Table 3 jpm-12-01013-t003:** Genetic information of the seven inherited rare variants identified in this study.

Gene and SNP	Mutation Location	Inheritance	Taiwan Biobank	ALFA	PROVEAN	SIFT	PolyPhen-2	Mutation Taster
*KLHL17* rs186429850	chr1:898542:A > G c.1096A > G p.T366A	Maternal	0.005137	0.000338	Neutral	Tolerated	Benign	Disease causing
*TDO2* rs183229581	chr4:156835551:T > A c.803T > A p.F268Y	Paternal	0.01	0	Neutral	Damaging	Probably damaging	Disease causing
*TRRAP* rs55755466	chr7:98574585:G > C c.8250G > C p.E2750D	Paternal	0.007581	0.000317	Neutral	Tolerated	Possibly damaging	Disease causing
*EIF3F* rs367735033	chr11:8008909:C > T c.10C > T p.P4S	Paternal	0.003309	0.000066	Neutral	Damaging	Damaging	Disease causing
*DICER1* rs772381832	chr14:95599687:T > A c.109A > T p.I37F	Maternal	0.000989	0.000008	Neutral	Damaing	Probably damaging	Disease causing
*ATP10A* rs184009994	chr15:25947187:C > T c.2636G > A p.R879H	Paternal	0.001648	0.000009	Deleterious	Damaging	Possibly damaging	Disease causing
*CDH15* rs149963083	chr16:89246698:G > A c.292G > A D98N	Paternal	0.001649	0.000171	Deleterious	Tolerated	Probably damaging	Disease causing

## Data Availability

Not applicable.

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
