# Peer review of "Two Genetic Mechanisms in Two Siblings with Intellectual Disability, Autism Spectrum Disorder, and Psychosis"

_jpm, 2022, doi:10.3390/jpm12061013_

Round 1

Reviewer 1 Report

The manuscript is very well writen and the logical is very clear to follow. Also, The authors using differnt sequenicng or experimetal approches to show the results and make the validadation. A minor comment regarding the figures should be showing in a higher resolution. 

Author Response

We have improved the resolution of all the figures in our revised manuscript, thanks for your suggestion.

Reviewer 2 Report

Genetic testing results of two siblings effected by ID and ASD are presented in this manuscript. CNV array and WGS were used to search for pathogenic variants, PCR and Sanger sequencing were used to validate variants. A microdeletion and some rare mutations were found in the two patients, respectively. A comprehensive discussion on these mutations is provided in the manuscript.

However, according to the "Instructions for Authors" provided by JPM editorial board, I think this manuscript should be submitted as a "case report", rather than an "article". As a case report, more details of data analysis and symptoms should be provided.

Author Response

Thanks for the review and comments on our manuscript.

We have added more clinical information about the family in our revised manuscript.

Regarding whether our manuscript should be considered as an article or case report, I would like to raise a recent paper for consideration. The paper entitled “Multiple rare inherited variants in a four generation schizophrenia family offer leads for complex mode of disease inheritance” was published in Schizophrenia Research (Schizophrenia Research 216 (2020) 288–294. The paper described a family with multiple members affected with schizophrenia. All the affected members shared a combination of multiple rare mutations inherited from their grandparents, who were unaffected carriers of these rare mutations. This finding unveiled a new model of schizophrenia genetics, which is insightful for understanding the genesis of schizophrenia.

Our report discovered two distinct genetic deficits in a family with two siblings affected with neuropsychiatric disorders. This finding is different from what we used to think about the genetics of affected sib-pairs, who usually share common genetic deficits. Also, we found that the genetic deficits of the elder son fit the oligogenic model of neuropsychiatric disorders, which adds new information to the genetic architecture of neuropsychiatric disorders. We think our findings bring new vision and insight into the complex genetics of neuropsychiatric disorders and contribute to the progress in this field.

Round 2

Reviewer 2 Report

Thanks for response and revision.

Rare mutations from unaffected parents may accumulated and caused the disorder together. This finding will change our strategy to conduct carrier screening to reduce rare diseases.

This finding indicate that we need a more complex model to evaluate how genes affect rare diseases. Another interesting and challenging topic.